# Cancer-Associated Fibroblasts Arising from Endothelial-to-Mesenchymal Transition: Induction Factors, Functional Roles, and Transcriptomic Evidence

**DOI:** 10.3390/biology14101403

**Published:** 2025-10-13

**Authors:** Junyeol Han, Eung-Gook Kim, Bo Yeon Kim, Nak-Kyun Soung

**Affiliations:** 1Chemical Biology Research Center, Korea Research Institute of Bioscience and Biotechnology, Cheongju 28116, Republic of Korea; jyhan@kribb.re.kr; 2Department of Biomolecular Science, Korea National University of Science and Technology, Daejeon 34113, Republic of Korea; 3Department of Biochemistry, Chungbuk National University College of Medicine, Cheongju 28644, Republic of Korea; egkim@chungbuk.ac.kr

**Keywords:** endothelial-to-mesenchymal transition (EndMT), cancer-associated fibroblasts (CAF), tumor microenvironment (TME), TGF-β signaling, single-cell RNA sequencing (scRNA-seq), spatial transcriptomics sequencing (ST-seq)

## Abstract

The tumor microenvironment, the surrounding milieu in which a tumor grows, promotes tumor proliferation and metastasis and enables immune evasion; among its key constituents is the cancer-associated fibroblast. To treat cancer and understand it more deeply, the study of cancer-associated fibroblasts is essential; these cells are known to arise from diverse sources, one of which is the vascular endothelium. Endothelial cells line the inner wall of blood vessels, and through endothelial-to-mesenchymal transition they lose their cobblestone morphology, tight intercellular adhesion, and endothelial functions, adopt a fibroblast-like spindle shape, loosen cell-to-cell contacts, and release abundant extracellular matrix components such as collagen and fibronectin. In this review, we describe the diverse factors that induce endothelial-to-mesenchymal transition and discuss how mesenchymalized endothelial cells influence tumor progression and the tumor microenvironment. We also present evidence from genome-wide profiling technologies for endothelial-to-mesenchymal transition-like cells across patient tumors and their impact on clinical outcomes.

## 1. Introduction

### 1.1. Cancer-Associated Fibroblasts (CAFs)

Cancer-associated fibroblasts (CAFs) are fibroblasts activated by cancer that help constitute the tumor microenvironment (TME) through extracellular matrix (ECM) production and remodeling, as well as the secretion of growth factors and cytokines [1]. CAFs have been reported to directly influence cancer biology—including tumor growth and metastasis [1,2]—and the proportion and activation state of CAFs correlate positively with tumor stage and recurrence, revealing a direct association between CAFs and clinical outcome [3,4]. A defining feature of CAFs is their marked heterogeneity: while many CAF populations promote tumor growth and dissemination, others can exert tumor-restraining functions (e.g., non-selective depletion of myofibroblastic CAFs in PDAC-aggravated disease and induced immunosuppression) [5,6]. One proposed source of this heterogeneity is their diverse cellular origins, which include tissue-resident fibroblasts, stellate cells (e.g., pancreatic stellate cells), bone marrow-derived mesenchymal stem cells, endothelial cells (ECs) undergoing endothelial-to-mesenchymal transition (EndMT), and adipocyte-to-fibroblast conversion [7,8,9,10].

### 1.2. Endothelial-to-Mesenchymal Transition (EndMT)

EndMT is a process by which endothelial cells (ECs) lose endothelial characteristics and acquire mesenchymal properties. It was first observed in the context of embryonic heart development in 1975, when Markwald and colleagues described, by electron microscopy and morphological analyses, the transformation of the primitive endocardium in the atrioventricular canal and outflow tract into cushion tissue (cardiac mesenchyme; EC-to-mesenchyme) [11]. Hallmark features of EndMT include a change from a cobblestone endothelial morphology to a spindle-shaped appearance, reduced expression of endothelial markers (e.g., VE-cadherin, CD31, vWF), and increased expression of mesenchymal markers (e.g., α-SMA, TAGLN, FN1, COL1A1/3) [12]. Decreases in junctional proteins such as CLDN5 are associated with heightened endothelial permeability and motility [13,14,15]. Canonical signaling axes implicated in EndMT include pathways mediated by TGF-β and IL-1β [12] (Figure 1).

Recent advances in bioinformatic technologies—including RNA sequencing (RNA-seq) and spatial transcriptomics sequencing (ST-seq)—have reshaped biological research [16,17,18]. In particular, single-cell RNA sequencing (scRNA-seq) has accelerated the identification of previously underappreciated cell types [19,20], and evidence supporting an EndMT-like endothelial cell state has likewise emerged [21].

EndMT has been linked to multiple diseases, including cardiac fibrosis [22], renal fibrosis [23], atherosclerosis [24], pulmonary arterial hypertension [25], cerebral cavernous malformations [26], and inflammatory bowel disease (IBD)-associated intestinal fibrosis [27]. Notably, the role of EndMT in cancer was first highlighted in 2007: ECs undergoing EndMT can give rise to CAFs that promote tumor angiogenesis and metastasis and are associated with poor patient outcomes [7,28]. Over the ensuing years, numerous studies have reported CAF-like cells of endothelial origin arising through endothelial-to-mesenchymal transition [7]; however, whereas some interpret cancer-induced transition as a CAF source, others emphasize endothelial barrier weakening and metastasis [14,29].

In this review, we systematically synthesize research on cancer-induced EndMT, covering the following: (1) factors that induce EndMT; (2) the influence of EndMT on cancer and the TME; (3) transcriptomics-based evidence for EndMT cell subtypes; and (4) how individual studies position EndMT-derived CAFs within tumor biology.

## 2. EndMT Induction in Cancer: Models and Factors

In this section, we introduce studies on the models and factors that induce EndMT in cancer, discuss the effects of EndMT on cancer and the TME, and summarize how individual studies position EndMT-derived CAFs (Figure 2).

### 2.1. Induction of EndMT Using Mouse Tumor Models

#### Zeisberg et al. (2007) [7]

Zeisberg et al. (2007) [7] provided the first experimental demonstration that cancer can induce EndMT. Using Tie2-Cre; Rosa26R-LacZ (R26R) reporter mice, they implanted B16F10 melanoma and, by combining β-gal (X-gal) staining with double immunostaining for mesenchymal markers (FSP1, α-SMA), identified lineage-traced CAFs arising from Tie2-expressing ECs. Cells co-expressing the endothelial marker CD31 and CAF markers—interpreted as transitional (partial EndMT) cells—were also detected: approximately 40% of FSP1^+^ cells and ~11% of α-SMA^+^ cells co-expressed CD31. In the RIP1-Tag2 spontaneous pancreatic islet tumor model, FSP1/CD31 co-expression was likewise observed, indicating that EndMT can occur during tumorigenesis. Mechanistically, FSP1^+^ cells in B16F10 tumor sections showed strong TGF-β1 immunoreactivity, and treatment of primary mouse lung endothelial cells (MLECs) with TGF-β1-induced spindle-like morphological changes, decreased CD31, and increased FSP1, consistent with TGF-β-driven EndMT. This study first established cancer-induced EndMT via endothelial lineage tracing and advanced the view that ECs can contribute to the CAF compartment. As an early report, however, it did not yet provide functional evidence for CAF activities such as tumor promotion, ECM remodeling, or immune modulation, while nevertheless laying essential groundwork by proposing the possibility of cancer-derived CAFs.

### 2.2. Induction of EndMT Using Cancer-Conditioned Medium (CM)

#### 2.2.1. Wojciech M. Ciszewski et al. (2017) [30]

Ciszewski et al. (2017) [30] used CM from colorectal cancer cell lines (LS180, LoVo, and SNAI1-overexpressing LS180-SNAIL) to show that greater tumor invasiveness was associated with EndMT features in HMEC-1: increased cell length (spindle-like morphology), stress-fiber remodeling, decreased endothelial markers (claudin, ZO-1), and increased mesenchymal markers (α-SMA, FSP1, N-cadherin) at the protein level, thereby demonstrating CM-induced EndMT. They identified TGF-β2 as the CM component responsible for EndMT and showed mechanistically that TGF-β2 promotes EndMT via the ILK–MMP9–MRTF axis. This study indicates that cancer-secreted TGF-β2 can reprogram integrin- and cytoskeleton-based signaling to drive EndMT-mediated CAF-like differentiation.

#### 2.2.2. Krizbai et al. (2015) [14]

Krizbai et al. (2015) [14] prepared activated conditioned medium (ACM) by heating CM from melanoma (B16/F10, A2058) and breast cancer (MCF-7, MDA-MB-231) cells at 80 °C for 10 min to activate latent TGF-β and applied it to rat brain endothelial cells (RBECs) to induce EndMT. Protein-level analyses showed decreased endothelial markers (claudin-5, VE-cadherin) and increased mesenchymal markers (fibronectin, α-SMA), accompanied by reduced transendothelial electrical resistance (TEER) and enhanced tumor–cell adhesion and transmigration. These changes were suppressed by the TGF-βR1 inhibitor SB-431542, confirming TGF-β dependence. The ROCK inhibitor Y-27632 attenuated TGF-β1-induced α-SMA upregulation, and immunofluorescence (IF) demonstrated nuclear translocation of SRF upon TGF-β1 treatment, indicating that α-SMA induction by TGF-β1 is ROCK dependent and strongly implicating the Rho–ROCK–SRF pathway. Similar phenomena were observed in HUVECs: ACM derived from breast cancer cells (MDA-MB-231, SK-BR-3) elicited EndMT in a TGF-β-dependent manner. This study demonstrated that cancer-cell CM contains latent TGF-β that when activated, can trigger EndMT in ECs, and it focused on increased metastasis via endothelial dysfunction rather than a CAF-centered interpretation.

#### 2.2.3. Valentin Platel et al. (2022) [31]

Platel et al. (2022) [31] treated HUVECs with CM from SK-MEL-28 melanoma cells and observed reduced tube formation (capillary length), increased motility, enhanced actin stress fibers, and—by flow cytometry—an increased proportion of vWF^+^/α-SMA^+^ double-positive cells, supporting partial EndMT. Mechanistically, CM exposure elevated intracellular reactive oxygen species (ROS) in ECs; this was attenuated by NOX1 and NOX2 inhibitors (ML171 and GSK-2795039, respectively), which also partially rescued the decrease in capillary length. Although the experiments did not directly demonstrate conversion of endothelium into bona fide CAFs, the introduction and discussion proposed EndMT as a contributor to CAF generation, thereby suggesting the plausibility of EC-to-CAF transition in this context.

#### 2.2.4. Clara Bourreau et al. (2025) [32]

Bourreau et al. (2025) [32] reported that treating HUVECs for 48–72 h with CM from non-small cell lung cancer (NSCLC) lines (NCI-H1755, A549, NCI-H23, NCI-H1437, NCI-H1975) elicited partial EndMT. Cells exhibited elongation with increased stress fibers, and CM from several lines also enhanced motility. By flow cytometry, some CM-treated conditions showed higher proportions of endothelial/mesenchymal marker-defined cells—vWF^−^/α-SMA^+^ (and CD31^−^/CD44^+^)—supporting EndMT features. Mass spectrometry-based secretome profiling of A549 and NCI-H1755 revealed significant enrichment of signaling pathways related to EndMT, angiogenesis, and reactive oxygen species, which ranked among the most dominant pathways. Factors previously implicated in EndMT, including SPP1 (osteopontin) [33] and SERPINE1 (PAI-1) [29], were increased in the secretome, reinforcing the plausibility of CM-driven EndMT. While the study did not provide direct evidence of endothelial conversion into bona fide CAFs, the discussion linked partial EndMT to an intermediate step toward CAF formation, suggesting the possibility of EndMT-derived CAFs.

### 2.3. Induction of EndMT via Co-Culture

Linghui Nie et al. (2014) [34]

Nie et al. (2014) [34] showed that co-culture of the esophageal adenocarcinoma (EAC) cell line OE33 with human esophageal microvascular endothelial cells (HEMECs), as well as treatment with cancer CM, induced hallmark EndMT features: spindle-like morphological changes with increased motility and reduced tube formation; decreased endothelial markers (CD31, VE-cadherin, vWF); and increased mesenchymal markers (FSP1, α-SMA, vimentin, desmin, COL1A2, Snail) at both protein and mRNA levels. Under co-culture, collagen gel contraction was enhanced—indicating myofibroblast-like function—and mRNA expression of the pro-inflammatory enzyme COX-2 increased, accompanied by nuclear translocation of NF-κB p65 and elevated mRNA levels of endothelial inflammatory activation markers (VCAM-1, ICAM-1), consistent with activation of pro-inflammatory pathways. Mechanistically, CM-induced EndMT was attenuated by siRNAs targeting IL-1β and TGF-β2, establishing their involvement. In EAC patient specimens, CD31/FSP1 double-positive EndMT cells were detected, predominantly at the invasive front, suggesting an association with tumor invasion. During cancer-induced EndMT, VEGF expression and secretion increased while VEGFR2 decreased, implying a paracrine role for secreted VEGF rather than an autocrine effect within ECs; VEGF co-expression in putative EndMT cells was also observed in patient tissues. This work demonstrates that co-culture and CM can drive EndMT and provides functional evidence consistent with CAF-like behavior—upregulation of ECM proteins, increased collagen contraction, and activation of inflammatory pathways—while further suggesting that cancer-induced EndMT may serve as a source of VEGF-secreting CAFs in EAC.

### 2.4. Induction of EndMT via 3D Modeling

#### 2.4.1. Se-Hyuk Kim et al. (2019) [35]

Kim et al. (2019) [35] induced EndMT using cancer cells cultured as three-dimensional (3D) spheroids rather than in two-dimensional (2D), either by applying spheroid-derived CM to HUVECs or by establishing 3D co-cultures of cancer cells with ECs. CM from NSCLC NCI-H460 cells grown as 3D spheroids more strongly decreased the endothelial marker CD31 and increased the mesenchymal marker α-SMA in HUVECs compared with 2D-derived CM, indicating a more potent induction of EndMT. Likewise, when NSCLC cell lines (NCI-H460, A549, SK-MES-1) were co-cultured with HUVECs under 2D versus 3D conditions, 3D co-culture nearly abolished endothelial markers (CD31, VE-cadherin) and significantly elevated the mesenchymal marker α-SMA, demonstrating that 3D interactions enhance EndMT. The 3D co-culture also increased spheroid compactness and stiffness, a phenomenon comparable to that observed in co-cultures of WI-38 human embryonic lung fibroblasts with cancer cells.

Mechanistically, 3D co-culture significantly increased the active-form readout of GSK-3β (p-Tyr216), with no change at the inhibitory Ser9 site. Pharmacologic inhibition of GSK-3β with CHIR99021 restored endothelial markers (CD31, VE-cadherin) and reduced the mesenchymal marker α-SMA, reversing EndMT features. Co-treatment with a β-catenin inhibitor MSAB did not negate the EndMT suppression by CHIR99021, indicating that GSK-3β-dependent control of EndMT in this setting is largely independent of canonical Wnt/β-catenin signaling. Functionally, 3D cancer–EC co-culture conferred resistance to the anticancer drug gefitinib; adding CHIR99021 increased apoptosis and mitigated this resistance. In mouse xenografts derived from 3D co-cultures of H460 cells and HUVECs, the combination of gefitinib and CHIR99021 reduced tumor size, decreased Ki-67-positive proliferating cells and CD31-positive endothelial cells, and lowered the fibrotic burden.

This study underscores the importance of physiologically relevant 3D systems for investigating EndMT. The increased spheroid compactness under 3D cancer–EC co-culture, together with in vivo reductions in fibrotic regions and modulation of drug resistance upon EndMT control, supports a functional role consistent with CAF-like behavior. The authors further noted the potential transition toward myofibroblast-like cells via EndMT, proposing EndMT-derived myofibroblast-like populations in this context.

#### 2.4.2. Ju Hun Yeon et al. (2018) [36]

Yeon et al. (2018) [36] employed a 3D microfluidic device that recapitulated the TME to induce EndMT. The device incorporated a type I collagen gel and a HUVEC monolayer to model ECM and endothelium; a hydrostatic height difference between the two side channels generated interstitial fluid flow (IFF). Tumor-derived exosomes from mouse B16BL6 melanoma were perfused through the channels to drive EndMT. The resulting cells were operationally defined as CAFs based on three criteria: morphological changes with active filopodia; increased mesenchymal markers (vimentin, FSP1); and decreased endothelial marker (VE-cadherin). In the presence of IFF, the number of such CAFs migrating from the endothelial monolayer into the collagen gel increased in an exosome dose- and time-dependent manner (rising through day 5, then gradually declining). Mechanistically, exosomal TGF-β mediated EndMT, as CAF formation was suppressed by a TGF-β-neutralizing antibody and a TGF-β receptor I kinase inhibitor SD-208. The authors also reproduced the reported effect of mesenchymal stem cell (MSC)-derived exosomes on suppressing myofibroblast differentiation [37] as inhibition of CAF formation within this device, supporting its use as an experimental platform for drug discovery targeting EndMT-derived CAFs. This study underscores the importance of 3D, microfluidic TME-mimetic systems for EndMT research and advances the view that CAFs can arise via EndMT. However, because CAFs were defined by morphology/marker criteria without direct functional verification (e.g., increased ECM deposition, collagen contraction, immunosuppression, or enhanced tumor progression), additional work using this 3D model to establish CAF functions is warranted.

### 2.5. EndMT Induced by Tumor Microenvironment (TME) Constituents

#### 2.5.1. Wen-Fei Wei et al. (2023) [29]

Wei et al. (2023) [29] demonstrated that EndMT can be driven not only by cancer cells but also by TME constituents—specifically, CAFs. CM from patient-derived CAFs of cervical squamous cell carcinoma (CSCC) induced spindle-like morphological changes, increased motility, decreased VE-cadherin, and increased mesenchymal markers (α-SMA, vimentin) in human lymphatic endothelial cells (HDLECs), consistent with EndMT. The CAF CM also promoted lymphangiogenesis and increased endothelial permeability. In CSCC specimens, CAF abundance (α-SMA staining) positively correlated with lymphatic vessel density (LYVE-1), and cases with metastasis exhibited a higher proportion of α-SMA^+^/LYVE-1^+^ co-expressing lymphatic vessels. In vivo, implantation of CM-treated HDLECs increased vessel length in mice, and footpad injections of the cervical cancer cell line SiHa with CM enhanced lymphatic invasion and the frequency of α-SMA^+^/LYVE-1^+^ lymphatic vessels.

Mechanistically, CAF-secreted PAI-1 activated ERK and AKT signaling in ECs via LRP1, thereby inducing EndMT; this was mitigated by a PAI-1-neutralizing antibody, siLRP1, the ERK pathway inhibitor U0126, and the AKT inhibitor MK-2206. In CSCC tissues, PAI-1 co-localized with α-SMA-positive CAFs and was significantly elevated in metastatic cases. PAI-1 levels correlated positively with lymphatic vessel density and with α-SMA^+^/LYVE-1^+^ vessels, and analysis of The Cancer Genome Atlas (TCGA) associated higher PAI-1 expression with poorer patient survival. This study establishes that CAFs within the TME can induce EndMT in ECs and, rather than arguing for conversion into CAFs, emphasizes altered endothelial behavior that contributes to increased lymphatic invasion and metastasis.

#### 2.5.2. Tze-Sing Huang Group (Chi-Shuan Fan et al., 2018 [33]; Fan et al., 2019 [38])

Macrophages have been proposed as another TME constituent capable of provoking EndMT. In 2018, the Huang group showed that treating HUVECs (and EC-RF24) with osteopontin (OPN)—which is highly expressed across multiple TME cell types including macrophages—induces EndMT [33]. Recombinant OPN reduced endothelial markers (VE-cadherin, Tie1/2, CD31), increased mesenchymal markers (α-SMA, fibronectin), diminished gap-junction activity, and enhanced cell migration and invasion. Mechanistically, OPN bound integrin αvβ3 to activate PI3K–AKT, suppressed TSC2, and thereby promoted mTORC1-dependent HIF-1α protein synthesis. HIF-1α then drove transcription of the EMT-associated factor TCF12, which together with an EZH2/HDAC complex repressed the VE-cadherin promoter and accelerated EndMT. Conditioned medium from OPN-induced EndMT ECs augmented colorectal cancer (CRC) progression, as evidenced by increased tumor growth and metastasis when applied to the CRC cell line HCT-115 both in vitro and in vivo. Microwestern array profiling of this conditioned medium showed elevated HSP90α, which expanded CD44^+^/CD326^+^ cancer-stem-like populations and increased CRC stemness.

The work was extended in 2019 to pancreatic ductal adenocarcinoma (PDAC) [38]. In mouse tumor-implantation experiments using Panc02 cancer cells and the mouse endothelial line 3B-11, co-injection of OPN-induced EndMT ECs increased tumor size and weight and enhanced infiltration by M2-type macrophages. Conditioned medium from these OPN-EndMT ECs also promoted macrophage polarization, lowering mRNA levels of M1-associated IL-1β and TNF-α while increasing M2-associated CD163, CD204, IL-10, and TGF-β. Mechanistically, EC-derived HSP90α acted on macrophages via TLR4 and CD91, engaging the MyD88–JAK2/TYK2–STAT3 pathway to upregulate Hsp90α transcripts and further HSP90α secretion within macrophages. In vivo, an anti-HSP90α antibody attenuated the EndMT-driven increases in Panc02 tumor burden and M2 macrophage infiltration, strengthening the causal role of HSP90α in this setting.

Notably, while these studies emphasized the close relationship between EndMT ECs and macrophages, they did not directly demonstrate OPN secretion by macrophages or show that macrophage-derived conditioned medium alone induces EndMT. Rather, they documented macrophage proximity to α-SMA^+^/CD31^+^ EndMT cells in CRC tissues and high OPN staining adjacent to EndMT regions.

In addition, OPN-induced EndMT in HUVECs and EC-RF24 was used to define an EndMT index based on pronounced downregulation of the long non-coding RNAs LOC340340, LOC101927256, and MNX1-AS1. When compared with TCGA PDAC data, EndMT-index-positive cases correlated with T4 stage, suggesting a link between EndMT and advanced disease with poorer prognosis.

Taken together, Fan et al., 2018 [33], suggested that OPN-induced EndMT ECs exert CAF-like functions such as promoting tumor growth and metastasis, and Fan et al., 2019 [38], went further to explicitly describe these OPN-EndMT ECs as EndMT-derived CAFs, demonstrating tumor-promoting and immune-modulatory roles.

### 2.6. Induction of EndMT via Targeted Gene Perturbation

#### 2.6.1. Roselyne Tournaire Group (Julie Garcia et al., 2012 [39]; Marjorie Adjuto-Saccone et al., 2021 [40])

The Roselyne Tournaire group reported in 2012 and 2021 that decreased TIE1 and TNF-α signaling can induce EndMT in endothelial cells and contribute to tumor stroma formation, particularly CAF accumulation.

In 2012, Julie Garcia et al. reduced TIE1 expression in human microvascular endothelial cells (HMVECs) using siRNA and observed spindle-like morphological changes, increased motility, reduced endothelial markers (CD31, VE-cadherin, CD34, FVIII) at mRNA and protein levels, and increased mesenchymal markers (α-SMA, S100A4, COL1A1, SM22α, N-cadherin), consistent with EndMT [39]. Mechanistically, TIE1 knockdown activated ERK1/2, ERK5, and AKT, which in turn activated the SNAI2 (Slug) promoter; Slug knockdown blunted EndMT induction. In human pancreatic tumor specimens, immunofluorescence identified cells co-expressing CD31 with mesenchymal markers (α-SMA, S100A4, SM22α, FAP, N-cadherin), indicating partial EndMT. Endothelial cells that did not co-express mesenchymal markers retained TIE1, whereas partial EndMT cells lacked TIE1 co-expression, suggesting an association between reduced TIE1 and EndMT in patients.

In 2021, Marjorie Adjuto-Saccone et al. showed that recombinant TNF-α in HMVECs induced spindle-like morphology, increased motility, suppression of angiogenesis, and EndMT-associated marker changes. TNF-α decreased TIE1 mRNA and protein, and TIE1 overexpression partially attenuated TNF-α-induced EndMT. TNF-α increased the mRNA and protein levels of EndMT-related transcription factors (SNAI1, SNAI2, ZEB2), and siRNA against these factors suppressed TNF-α-driven EndMT. TNF-α also activated PI3K/AKT, ERK1/2, ERK5, JNK, and NF-κB signaling, although the study did not directly link each pathway to specific EndMT outputs. Secretome analysis by mass spectrometry showed increased secretion of pro-inflammatory proteins and COL1A1 after TNF-α treatment, and Ingenuity pathway analysis indicated activation of pathways associated with inflammation, fibrosis, migration, and pro-tumoral programs. These features align closely with CAF functions, and the authors proposed TNF-α-induced EndMT as evidence supporting CAF-like conversion. In vivo, systemic TNF-α administration in a PDAC syngeneic mouse model increased the fibrotic region and α-SMA^+^ cells within tumors, suggesting that TNF-α elevates CAF abundance in vivo.

Reversibility was also demonstrated. Upon withdrawal of TNF-α from EndMT-induced cells, endothelial morphology and tube-forming capacity recovered, CD31 increased, and α-SMA decreased.

Collectively, Julie Garcia et al., 2012 [39], proposed that EndMT driven by reduced TIE1 could serve as a source of CAFs. Building on this, Marjorie Adjuto-Saccone et al., 2021 [40], showed that TNF-α accompanied by TIE1 downregulation, induces EndMT with increased secretion of pro-inflammatory proteins and collagen and promotes tumor-supportive traits, including enhanced fibrosis, thereby strengthening the evidence for EndMT-derived CAFs.

#### 2.6.2. Seo-Hyun Choi Group (Seo-Hyun Choi et al., 2016 [41]; 2018 [28])

In 2016, Seo-Hyun Choi et al. reported that loss of HSPB1 induces EndMT. In human pulmonary microvascular endothelial cells (HPMECs), siRNA-mediated HSPB1 knockdown decreased endothelial markers (VE-cadherin, CD31) and increased the mesenchymal marker α-SMA at both protein and mRNA levels, and it suppressed tube formation induced by recombinant VEGF. Overexpression of HSPB1 attenuated TGF-β1-induced EndMT and also suppressed radiation-induced EndMT, a phenomenon that the authors had previously established [42]. In vivo, in LSL-KrasG12D; Trp53flox/flox (KP) mice, endothelial HSPB1 knockdown via shRNA increased α-SMA^+^/CD31^+^ partial EndMT cells, and around tumor vessels, collagen deposition and expression of TGF-β1 and FSP1 were elevated. Tumor size and the number of PCNA^+^ proliferating cells also increased. In human lung cancer tissues, HSPB1^−^/CD31^+^ tumor endothelial cells frequently co-expressed α-SMA, indicating EndMT. Whereas the co-expression of HSPB1 and CD31 was prominent in non-fibrotic regions, vessels co-expressing HSPB1 and CD31 were scarce in fibrotic regions, suggesting that HSPB1 deficiency may contribute to tumor fibrosis.

This work presented functional evidence linked to CAF biology—CAF-associated markers, increased collagen deposition, heightened tumor fibrosis, and enhanced tumor growth—and explicitly proposed that EndMT caused by HSPB1 deficiency could serve as a source of CAFs.

The group subsequently showed in 2018 that radiation therapy in lung cancer can elicit EndMT and thereby influence recurrence, pericyte redistribution, activation of CD44v6^+^ cancer stem cell-like populations, and macrophage polarization. Endothelial p53 reduction mitigated these radiation-driven effects: in EC-p53KD (Tie2-Cre;Trp53flox/+) and EC-p53KO (Tie2-Cre;Trp53flox/flox) mice bearing KP tumors and exposed to irradiation, the increases seen in wild type—α-SMA^+^/CD31^+^ partial EndMT, collagen deposition, tumor size, and the number of metastatic nodules—were markedly reduced. Radiation-induced α-SMA^+^NG2^+^ pericytes were also diminished in EC-p53KO mice, yielding decreased pericyte coverage, increased vascular leak, and exacerbated hypoxia; consequently, the post-irradiation expansion of CD44v6^+^ CSC-like cells was hindered. Radiation elevated OPN mRNA and secretion in ECs, and OPN activated dormant CD44v6^+^ CSC-like cells under hypoxia; endothelial p53 knockout curtailed OPN secretion and blocked CSC activation. Radiation further promoted differentiation of Arg1^+^F4/80^+^ tumor-associated macrophages (M2-type), which was suppressed in EC-p53KO mice. Similar patterns were observed in irradiated patient samples, with higher proportions of α-SMA^+^/CD31^+^ EndMT vessels, NG2^+^/α-SMA^+^ pericytes surrounding vessels, increased CD44v6^+^ CSC-like cells, and SDF1^+^/CD206^+^/CD68^+^ M2-type macrophages.

Beyond p53, the study also examined TGFBR2 as a target. Unlike p53, endothelial TGFBR2 knockdown promoted EndMT: in EC-TGFBR2KD (Tie2-Cre;Tgfbr2flox/+) mice bearing tumors, irradiation led to greater tumor size and more metastatic nodules than in wild type, along with increased fractions of α-SMA^+^/CD31^+^ EndMT vessels and α-SMA^+^NG2^+^ pericytes. These increases were diminished in EC-p53KD/KO;TGFBR2KD mice, indicating that the effects of TGFBR2 knockdown are modulated by endothelial p53 status. However, given that TGF-β signaling is a principal driver of EndMT, the increase in EndMT observed with TGFBR2 reduction may seem paradoxical. In this study, the authors showed—both in vitro and in vivo—that endothelial cell-specific Tgfbr2 knockdown led to increased irradiation-induced p-SMAD2/3, and they proposed that loss of TGFBR2 can compensatorily amplify SMAD2/3 signaling via TGFBR1. Accordingly, even with reduced TGFBR2, the TGF-β pathway remains sufficiently activated to drive EndMT.

Using EC-tdTomato and EC-tdTomato;p53KO mice (Tie2-Cre;tdTomato and Tie2-Cre;tdTomato;Trp53flox/flox), the authors lineage-traced ECs to verify EndMT in vivo. While the work highlighted irradiation-induced EndMT as a regulator of the TME—including pericytes and macrophages—with potential prognostic impact, it did not claim conversion into CAFs, focusing instead on endothelial reprogramming within tumors.

### 2.7. Virus-Induced EndMT

#### Paola Gasperini et al. (2012) [43]

Gasperini et al. (2012) [43] showed that Kaposi’s sarcoma-associated herpes virus (KSHV) can trigger EndMT. Infection of neonatal human dermal microvascular endothelial cells (DMVECs) led to actin reorganization, increased motility, reduced endothelial markers (CD31, VE-cadherin, Tie2, CD34), and increased mesenchymal markers (CD146, NG2, vimentin, PDGFRβ, α-SMA). In Kaposi’s sarcoma (KS) specimens, LANA-positive cells exhibited low CD31 with high expression of mesenchymal markers (SMA, PDGFRβ, CD146), consistent with EndMT in infected cells.

Mechanistically, KSHV activated NOTCH signaling. After infection, mRNA levels of NOTCH targets HEY1 and HEY2 and EMT-associated transcription factors (Slug, ZEB1, ZEB2) increased, while treatment with a NOTCH inhibitor DAPT suppressed these changes and reduced mesenchymal marker induction, thereby inhibiting EndMT. Co-localization of LANA with HEY2 or ZEB1 in patient tissues further supported NOTCH involvement in KSHV-induced EndMT.

This study demonstrates that in virus-driven cancers, a tumor virus can induce EndMT in endothelial cells. The authors were cautious about CAF conversion, proposing that KSHV-mediated inactivation of glycogen synthase kinase-3 may prevent full differentiation into mature fibroblasts.

The inducers and models of cancer-induced EndMT, its impacts on cancer and the TME, and each study’s stance on cancer-derived CAFs—for the studies introduced in Section 2, “EndMT Induction in Cancer: Models and Factors”—are summarized in Table 1; the full dataset is provided in Appendix A.

## 3. EndMT-CAFs Revealed by Bioinformatics Tools

Advances in single-cell RNA sequencing (scRNA-seq) and related computational approaches have substantially strengthened evidence for EndMT cell states in patients [21,44]. Here, we introduce studies that leveraged scRNA-seq, bulk RNA-seq, spatial transcriptomics sequencing (ST-seq), and label-free quantitation (LFQ) proteomics to elucidate cancer-induced EndMT and its roles within cancer and the TME.

### 3.1. Han Luo et al. (2022) [44]

Luo et al. (2022) [44] analyzed scRNA-seq data comprising 855,271 cells across 10 solid cancer types to define CAF populations. Among CAF subtypes, they annotated a cluster co-expressing endothelial markers (e.g., vWF, PLVAP) and mesenchymal markers (e.g., ACTA2, RGS5) as CAFEndMT. This population was broadly present across cancers and was enriched in tumor and adjacent tissues relative to normal. Trajectory analysis suggested a differentiation path from tumor endothelial cells (TECs) to CAFEndMT and onward to cancer-associated myofibroblasts (CAFmyos). Compared with TECs and CAFmyos, CAFEndMT displayed a higher angiogenesis hallmark signature. Notably, endothelial cell-specific molecule 1 (ESM1) was highly expressed in CAFEndMT and was proposed as a putative CAFEndMT marker.

Using TCGA patient cohorts, a CAFEndMT gene-set signature associated with worse survival in several cancers (e.g., breast, gastric, colorectal), suggesting a link between CAFEndMT abundance and poor prognosis. CellPhoneDB-based analyses predicted strong ligand–receptor interactions between CAFEndMT and tumor-associated macrophages (TAMs); in NicheNet analyses, CAFEndMT CD44 and TAM SPP1 (osteopontin; OPN)—an EndMT-inducing factor [33]—ranked among the top interactions. Immunostaining in anaplastic thyroid, colorectal, and gastric cancers showed enrichment of SPP1^+^ TAMs within regions dense for CD44^+^/CD31^+^ CAFEndMT; spatial quantification indicated that intercellular distances ≤ 20 μm occurred significantly more frequently than expected by chance. In spatial transcriptomics from seven colorectal tumors, CAFEndMT and SPP1^+^ TAM signature enrichments were positively correlated (R = 0.23, *p* < 0.001), supporting close crosstalk between these populations.

Collectively, this work cataloged CAF subtypes across ten cancers, defined EndMT as a CAF subtype, and—based on trajectory analysis—proposed a continuum in which tumor endothelial cells (TECs) transition through CAFEndMT and subsequently progress to cancer-associated myofibroblasts (CAFmyos), thereby nominating EndMT as a plausible origin for a subset of CAFs.

### 3.2. Quanzhong Liu et al. (2025) [21]

Quanzhong Liu et al. (2025) [21] proposed an EndMT-like endothelial subtype termed COL1A1^+^ ECs in gastric cancer (GC) and linked this state to disease behavior and patient prognosis. From 72 GC scRNA-seq samples, cells were filtered for PECAM1^+^ profiles, yielding 9402 endothelial cells. Principal component analysis (PCA) and clustering identified 11 endothelial clusters. One cluster showed high expression of mesenchymal markers (ACTA2, COL1A1, RGS5) and enrichment of matrix-related pathway gene sets; this cluster was designated COL1A1^+^ ECs as an EndMT-consistent population. Differentially expressed genes (DEGs) elevated in COL1A1^+^ ECs relative to other clusters were compiled into an EndMT Signature (EdMTS).

EdMTS was higher in tumors than in adjacent normal tissues and was increased in advanced stages (T3–T4) compared with early stage (T1), indicating a progressive role of EndMT. EdMTS also varied by metastatic site, being higher in liver metastases than in peritoneal or ovarian lesions. In TCGA-STAD cohorts, EdMTS correlated positively with hypoxia, invasion, TGF-β signaling, and migration programs. Patients not responding to immunotherapy exhibited higher EdMTS; in the high-EdMTS group, genes associated with cytokine signaling, antigen processing, and TGF-β pathways were upregulated, and the proportion of PD-L1-positive ECs was increased—together suggesting a contribution of EndMT to immune evasion. Across TCGA-STAD and additional public datasets, high EdMTS was associated with poorer overall survival.

Pseudotime analysis positioned COL1A1^+^ ECs toward the terminal end of an endothelial trajectory; Branch Expression Analysis Modeling (BEAM) analysis identified genes enriched at this terminus, highlighting ECM-related pathways and mesenchymal features (COL1A1, ACTA2, TAGLN) consistent with CAF-like characteristics. Mechanistically, DoRothEA prioritized CEBPB as a key transcription factor for EndMT in this context. In vitro, CEBPB overexpression in HUVECs increased COL1A1 protein, and in TCGA-STAD data, COL1A1 and CEBPB were positively correlated, supporting a CEBPB-driven EndMT program in COL1A1^+^ ECs.

Intercellular crosstalk analyses using CellChat and tissue staining implicated the ANGPTL4–SDC4 axis in interactions between COL1A1^+^ ECs and tumor cells. ANGPTL4, expressed by ECs, has been associated with tumor angiogenesis and metastasis, whereas SDC4, expressed by cancer cells, is upregulated in GC and linked to invasion. In vitro, COL1A1 overexpression in HUVECs increased ANGPTL4, and co-culture with the gastric adenocarcinoma line MKN45 enhanced cancer cell invasion and proliferation, supporting a model in which ANGPTL4 from COL1A1^+^ ECs engages SDC4 on tumor cells to promote GC progression and spread.

Overall, by defining COL1A1^+^ ECs as an EndMT-featured endothelial state in GC and connecting EdMTS to metastasis patterns, immune dysregulation, and adverse prognosis, this study provided patient-level evidence that EndMT-like endothelial remodeling contributes to tumor progression. While the authors did not explicitly designate COL1A1^+^ ECs as EndMT-derived CAFs, the functional parallels—tumor promotion, metastatic association, immune pathways, and worse outcomes—support the concept and its relevance to CAF biology.

### 3.3. Minghui Hou et al. (2025) [45]

In gallbladder cancer (GBC), Minghui Hou et al. (2025) [45] delineated an EndMT-associated endothelial subtype, CD34^+^CD90^+^ ECs, and demonstrated its relevance to tumor progression and patient prognosis. Using scRNA-seq on tumor tissues from 12 treatment-naïve GBC patients (GBC1–12) and benign gallbladder tissues from six chronic cholecystitis patients (CC1–6), the authors isolated 8897 endothelial cells (endothelial markers: CD34, PECAM1, VWF) and subclustered them into eight endothelial subtypes. Among these, a scar-associated endothelial (SAEndo2) cluster characterized by selective expression of the mesenchymal marker CD90 was markedly enriched in GBC tumors compared with benign tissue. Differential expression profiling of SAEndo2 revealed upregulation of pathways related to ECM remodeling and EMT/mesenchymal programs. When CD34^+^CD90^+^ ECs were isolated from GBC tumors, they exhibited higher mRNA levels of mesenchymal markers (FN1, α-SMA, COL4A2) and EndMT-related transcription factors (TWIST, SLUG, SNAI) than CD34^+^CD90^−^ ECs, together with spindle-like morphology. Immunofluorescence further showed stronger signals for mesenchymal/ECM markers (MMP-9, FN1, PDGF, collagen) and phospho-SMAD2/SMAD3 (indicative of TGF-β/Smad activity) in CD34^+^CD90^+^ ECs, supporting an EndMT-linked phenotype.

Clinically, higher intratumoral counts of CD34^+^CD90^+^ ECs were associated with worse prognosis, and analysis of the European Genome-phenome Archive (EGA)-GBC bulk RNA-seq data showed lower survival in cases with elevated SAEndo2 scores. Functionally, CD34^+^CD90^+^ ECs promoted the metastatic behavior of the GBC-SD cell line: in co-culture, they enhanced migration and invasion more than CD34^+^CD90^−^ ECs, effects that were suppressed by TGF-β pathway inhibitors (ITD and PFD). CellChat analysis likewise indicated significantly heightened TGF-β signaling between SAEndo2 and tumor cells. In vivo metastasis models mirrored these findings, with greater metastatic burden in mice receiving GBC-SD together with CD34^+^CD90^+^ ECs compared with CD34^+^CD90^−^ ECs.

Beyond GBC, the same endothelial state was detected in hepatocellular carcinoma (HCC). Re-analysis of public data (GSE149614) showed a higher proportion of CD34^+^CD90^+^ ECs in HCC tumors than in matched normal tissues, and patient-sample immunofluorescence confirmed their presence. Pathway analysis again highlighted ECM-related programs, paralleling the GBC results.

This study isolated endothelial cells from gallbladder cancer (GBC) and identified the SAEndo2 (CD34^+^CD90^+^ EC) subset. Although CD34^+^CD90^+^ ECs are endothelial, they exhibited numerous mesenchymal-associated features, promoted tumor metastasis, and were associated with poor patient prognosis, suggesting the possibility of a link to CAF biology. However, the authors did not explicitly claim that CD34^+^CD90^+^ ECs are EndMT-derived CAFs, and they noted the need for further study regarding the relationship between SAEndo2 and CAFs.

### 3.4. Li Ji et al. (2025) [46]

Li Ji et al. (2025) [46] combined CM-based EndMT induction assays with spatial transcriptomics sequencing (ST-seq), RNA-seq, and label-free quantitation (LFQ) proteomics to corroborate that drug resistance in glioblastoma (GBM) can induce EndMT and may further give rise to CAFs. First, ST-seq was used to analyze tumor tissues from patients with primary GBM (GBM) and recurrent GBM (rGBM). Among fibroblast subclusters, the cluster that in rGBM showed high ECM Gene Ontology enrichment and elevated expression of CAF markers (COL3A1, COL1A1, COL1A2, FN1) was defined as the CAF-enriched cluster. This cluster occupied a much larger proportion in rGBM than in GBM. Using the Chinese Glioma Genome Atlas (CGGA) and TCGA, gene set enrichment analysis (GSEA) showed that the CAF-enriched cluster gene set positively correlated with a multidrug-resistance gene set, and patients with higher CAF gene-set scores exhibited poorer survival.

Building on the authors’ earlier work implicating EndMT-derived CAFs in temozolomide (TMZ) resistance in GBM [47], this study tested causality using CM. CM from TMZ-sensitive (TMZ-S) or TMZ-resistant (TMZ-R) GBM cells was applied to human cerebral microvascular endothelial cells (HCMECs). CM from TMZ-R GBM decreased VE-cadherin and increased α-SMA, indicating EndMT. When HCMECs were co-cultured with TMZ-R versus TMZ-S GBM, TGF-β and ECM proteins (COL1A1, FN1) increased, again indicating EndMT. Bulk RNA-seq and LFQ proteomics on HCMECs co-cultured with TMZ-R versus TMZ-S GBM identified differentially expressed genes (DEGs) and differentially secreted proteins (DSPs); among seven commonly upregulated factors, TNC and FLNC were selected as targets because they correlated positively with ECM proteins (COL1A1, COL1A2, FN1) and with CAF infiltration. Protein levels of TNC and FLNC were also elevated in HCMECs co-cultured with TMZ-R GBM. Treating HCMECs with recombinant TNC and FLNC increased TGF-β, COL1A1, and FN1; similarly, treating HCMECs with the chemotherapeutic etoposide (ETO) increased TGF-β, COL1A1, FN1, and TNC/FLNC, strongly suggesting that drug resistance can induce EndMT via TNC and FLNC.

To explore therapeutic reversal, candidate compounds predicted to bind active sites of TNC and FLNC were screened. Among these, punicalin (PNC), applied together with TMZ-R GBM CM to HCMECs, suppressed CM-induced upregulation of TNC and FLNC and blocked the increase in ECM proteins (COL1A1, FN1) and α-SMA, as well as the decrease in endothelial markers (VE-cadherin, CD31). In a mouse xenograft model using TMZ-R GBM cells, the combination of TMZ and PNC overcame TMZ resistance and reduced tumor size and weight, whereas TMZ or PNC alone showed no effect. These findings suggest that targeting TNC and FLNC can inhibit EndMT and help overcome drug resistance both in vitro and in vivo.

Overall, the study links recurrence and chemoresistance in GBM to EndMT-derived CAFs. Using ST-seq and bulk RNA-seq, it associated patient survival and drug resistance with CAF features, and via TMZ-R GBM CM-induced EndMT experiments, it suggested that EndMT may be a source of CAFs in rGBM. Notably, while the authors consistently argued for EndMT-derived CAFs, the evidence for EndMT relied mainly on marker changes in HCMECs and increases in drug resistance; likewise, within the ST-seq-defined CAF-enriched cluster, endothelial markers were not demonstrated, and a distinct EndMT-enriched cluster was not separately defined.

For the studies introduced in Section 3, “EndMT-CAFs Revealed by Bioinformatics Tools,” we summarized those that demonstrated cancer-induced EndMT using transcriptomic and proteomic technologies, covering cluster-annotation markers and key phenotypes (Table 2); the full dataset, including mechanisms and CAF-origin claims, is available in Appendix A.

## 4. Conclusions

In this review, we summarized studies reporting that EndMT in ECs is counted among the origins of CAFs and described its impact on cancer and the TME. We documented induction of EndMT in diverse cancers—including melanoma, colorectal cancer, breast cancer, esophageal adenocarcinoma, lung cancer, cervical squamous cell carcinoma, pancreatic cancer, Kaposi’s sarcoma, gallbladder cancer, gastric cancer, hepatocellular carcinoma, and glioblastoma—and outlined induction approaches such as mouse tumor models (transplantation and spontaneous), treatment with conditioned medium (CM) from cancer cells and CAFs, co-culture, targeted gene perturbation, ligand stimulation, exosome exposure, irradiation, viral infection, and 3D culture systems. Key phenotypes of EndMT included spindle-like morphology, impaired angiogenesis and barrier function, and reduced EC markers (attenuation of endothelial traits), together with increased motility, elevated secretion of ECM proteins and immune-related factors, and upregulation of mesenchymal markers (acquisition of mesenchymal features). Additionally, studies using rapidly advancing bioinformatics technologies (e.g., scRNA-seq, ST-seq, bulk RNA-seq, and label-free quantitation [LFQ] proteomics) have also provided corroborative evidence for EndMT-derived CAFs. Mechanistically, the reported drivers of EndMT include TGF-β signaling, cytoskeletal/mechanical stress, ROS, OPN, PAI-1, IL-1β, GSK-3β, HSP90α, TIE1, TNF-α, HSPB1, and NOTCH signaling. Notably, TGF-β signaling has been recurrently observed across multiple studies and models, underscoring its central role in EndMT induction. In addition, factors such as OPN, PAI-1, and ESM1 have been repeatedly reported, suggesting that they represent important targets in EndMT biology. In particular, OPN has been implicated in EndMT across diverse cancer types and is highly expressed by multiple cell populations within the TME; taken together, these observations nominate OPN as a potentially critical target for future investigations of cancer-associated EndMT. In cancer, EndMT was closely associated with tumor growth and metastasis and showed interactions with neighboring cells, including pericyte redistribution and promotion of macrophage differentiation. Several studies, in particular, linked EndMT to adverse outcomes such as recurrence and drug resistance, highlighting the need to develop therapeutics that target EndMT in addition to tumor cells. Additionally, the EndMT-derived CAFs highlighted in this review exhibited pro-tumor characteristics. Given the heterogeneity of CAF subtypes—some of which can be pro- or anti-tumor—EndMT-derived CAFs to date appear to align predominantly with pro-tumor features, while exceptions cannot be excluded (Figure 3).

Among the papers reviewed here, some explicitly argued that cancer-induced EndMT is a source of CAFs, whereas others presented only the possibility or focused on endothelial perspectives such as angiogenesis and barrier weakening. To distinguish EndMT-derived CAF differentiation from mere endothelial dysfunction, further evidence beyond endothelial-to-mesenchymal state change—specifically, demonstrations of CAF-typical functions such as increased ECM secretion, participation in matrix contraction, and heightened release of immune-related factors—would provide a more rigorous foundation.

## Figures and Tables

**Figure 1 biology-14-01403-f001:**
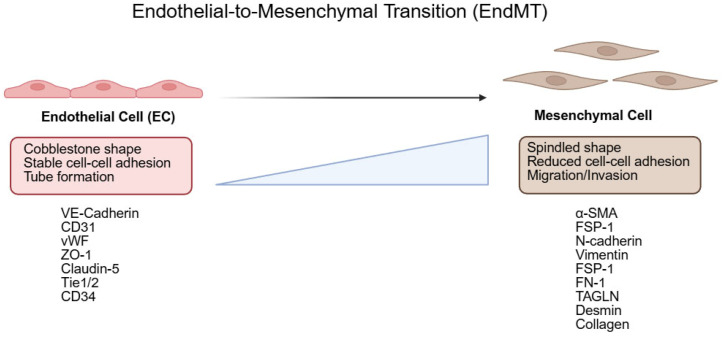
Hallmark features of endothelial-to-mesenchymal transition (EndMT). Schematic summarizing EndMT: endothelial cells (ECs) lose cobblestone morphology, stable cell–cell adhesion, and tube-forming capacity, with decreased endothelial markers (VE-cadherin, CD31, vWF, ZO-1, claudin-5, Tie1/2, CD34), while mesenchymal traits emerge—spindled morphology, reduced cell–cell adhesion, migration/invasion—and mesenchymal markers increase (α-SMA, FSP1, N-cadherin, vimentin, FSP-1, FN1, TAGLN, desmin, collagen). Created in BioRender. HAN, J. (2025) https://BioRender.com/et0cpj8 (accessed on 19 September 2025).

**Figure 2 biology-14-01403-f002:**
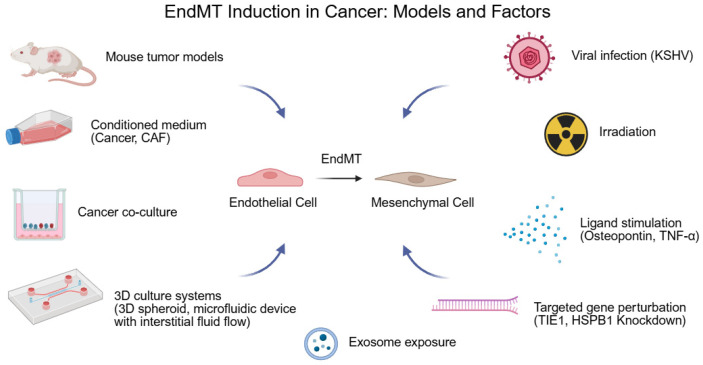
Endothelial-to-mesenchymal transition (EndMT) induction in cancer: models and factors. This schematic summarizes the experimental models and inducing factors that trigger EndMT in cancer. Reported approaches include mouse tumor models (transplantation and spontaneous), conditioned medium (CM) from cancer cells or cancer-associated fibroblasts (CAFs), cancer co-culture systems, 3D culture systems (3D spheroids; microfluidic devices with interstitial fluid flow), exosome exposure, targeted gene perturbations (Tie1, HSPB1 knockdown), ligand stimulation (osteopontin, TNF-α), irradiation, and Kaposi’s sarcoma-associated herpesvirus (KSHV) infection. Created in BioRender. HAN, J. (2025) https://BioRender.com/njodwu9 (accessed on 19 September 2025).

**Figure 3 biology-14-01403-f003:**
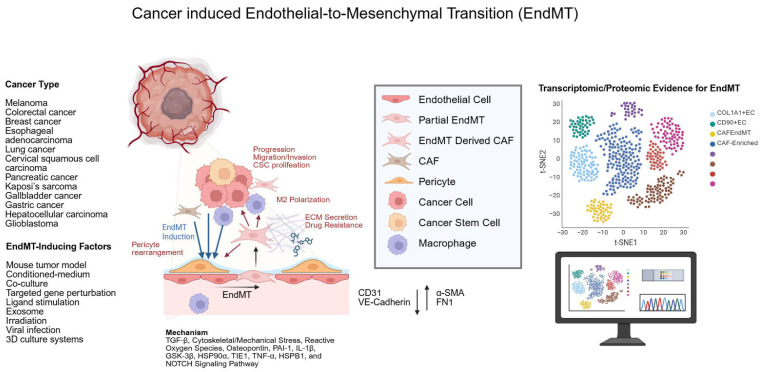
Schematic summary of cancer-induced endothelial-to-mesenchymal transition (EndMT) and its contribution to cancer-associated fibroblast (CAF) origins. EndMT has been reported across diverse cancers. EndMT is triggered by various induction factors, including but not limited to in vivo models and experimental perturbations. Hallmark EndMT phenotypes include loss of endothelial markers (e.g., CD31 and VE-cadherin), gain of mesenchymal markers (e.g., α-SMA and FN1), spindle-like morphology, impaired cell–cell adhesion, and enhanced secretion of ECM and immune-related factors. Mechanistically, diverse signaling pathways, including the TGF-β signaling pathway, are involved. Interactions with surrounding cells include pericyte redistribution, macrophage type 2 polarization (M2 polarization), and cross-talk with cancer and cancer stem cells (CSC), contributing to progression, migration, invasion, and drug resistance. Transcriptomic analyses (single-cell RNA-sequencing, spatial transcriptomics sequencing, bulk RNA-sequencing) and label-free quantitation (LFQ) proteomics provide corroborative evidence for EndMT-derived CAFs. In the schematic, blue arrows denote the influences of cancer and neighboring cells on endothelial cells (ECs), whereas red arrows denote the influences of EndMT-derived CAFs on cancer and neighboring cells within the tumor microenvironment (TME). Created in BioRender. HAN, J. (2025) https://BioRender.com/m2wtt5m (accessed on 19 September 2025).

**Table 1 biology-14-01403-t001:** Summary of cancer-induced EndMT (condensed): inducers and impacts on cancer/TME. Columns included: model of induction; induction protocol; EndMT key phenotype (including impacts on cancer and the TME). The full dataset (including Reference, Cancer type, EC type, Mechanism, and EndMT-derived CAF claim) is provided in Appendix A.

Reference	Model of Induction	Induction Protocol	EndMT Key Phenotype
Zeisberg et al., 2007 [7]	Tumor implantation, spontaneous tumor induction	Subcutaneous implantation of B16F10 in C57BL/6; spontaneous tumors in Rip1-Tag2 transgenic mice	β-gal^+^/FSP1^+^ and β-gal^+^/α-SMA^+^ traced cells, CD31^+^/FSP1^+^ and CD31^+^/α-SMA^+^ cells (partial EndMT), FSP1^+^/TGF-β1^+^ cells
Ciszewski et al., 2017 [30]	Cancer CM	CM from colon cancer cells after 72 h culture mixed with EC medium at 1:2 and applied to HMEC-1 for 216 h	Spindle-like morphology; stress fibers increased; endothelial markers decreased (claudin-5, ZO-1); mesenchymal markers increased (α-SMA, FSP-1, N-cadherin)
Krizbai et al., 2015 [14]	Cancer CM	ACM (latent TGF-β activated by heating 80 °C, 10 min) treated on ECs for 48 h	Endothelial markers decreased (VE-cadherin, claudin-5); mesenchymal markers increased (fibronectin, β1-integrin, calponin, α-SMA); TEER decreased
Platel et al., 2022 [31]	Cancer CM	CM (72 h culture) mixed 1:1 with EC medium and treated for 24, 48, 72 h	Tube formation decreased; migration increased; actin stress fibers increased; vWF^+^/α-SMA^+^ double-positive cells increased (partial EndMT)
Bourreau et al., 2025 [32]	Cancer CM	CM (72 h) mixed 1:1 with EC medium; treated for 48–72 h	Spindle-like morphology; stress fibers increased; loss of endothelial marker vWF; gain of mesenchymal markers α-SMA and CD44 (increased vWF^−^/α-SMA^+^ and CD31^−^/CD44^+^ cells); secretome showed enrichment of EndMT, angiogenesis, and ROS-related pathways
Nie et al., 2014 [34]	Cancer co-culture, cancer CM	Transwell co-culture with OE33 for 3, 6, 10 days; CM (24 h cancer culture in EC medium) applied to HEMEC for 6 days	Endothelial markers decreased (CD31, VE-cadherin, vWF); mesenchymal markers increased (FSP1, α-SMA, vimentin, desmin, COL1A2, Snail); migration increased; tube formation decreased; collagen gel contraction increased; CD31^+^/FSP1^+^ cells in EAC tissue; VEGF secretion increased with VEGFR2 decreased
Se-Hyuk Kim et al., 2019 [35]	3D spheroid co-culture, 3D-culture cancer CM	CM from 3D spheroids (3 days) applied to HUVECs for 24–48 h; ultra-low-attachment 3D co-culture for 3 days	Endothelial markers decreased (CD31, VE-cadherin); mesenchymal marker α-SMA increased; spheroid compactness increased; in vivo with CHIR99021: fibrosis and CD31 signal decreased; drug resistance mitigated
Ju Hun Yeon et al., 2018 [36]	3D microfluidic device with IFF, cancer exosomes	IFF generated by reservoir height difference; exosomes (1, 10, 50 μg/mL) for 1–7 days (from B16BL6-bearing C57BL/6J mice)	Morphological change with active filopodia; mesenchymal markers increased (vimentin, FSP-1); endothelial marker decreased (VE-cadherin)
Wen-Fei Wei et al., 2023 [29]	CAF CM	CM from patient-derived CAFs cultured 48 h in EC medium, applied to HDLEC for 24 h	Endothelial marker decreased (VE-cadherin); mesenchymal markers increased (α-SMA, vimentin); α-SMA^+^/LYVE-1^+^ lymphatic vessels increased; transendothelial migration increased
Chi-Shuan Fan et al., 2018 [33]	OPN treatment	Recombinant OPN 0.3 μg/mL for 15–24 h	Endothelial markers decreased (VE-cadherin, Tie1, Tie2, CD31); mesenchymal markers increased (α-SMA, fibronectin); migration and invasion increased; gap-junction activity decreased; cancer growth and metastasis increased; CD44^+^/CD326^+^ stemness population increased
Chi-Shuan Fan et al., 2019 [38]	OPN treatment	OPN 0.3 μg/mL for 24 h	Endothelial markers decreased (VE-cadherin, Tie1, Tie2, CD31); mesenchymal markers increased (α-SMA, fibronectin); lncRNA (LOC340340, LOC101927256, MNX1-AS1) decreased; tumor growth increased; M2 macrophage infiltration increased
Julie Garcia et al., 2012 [39]	Tie1 expression modulation	Tie1 siRNA knockdown for 48–72 h	Spindle-like morphology; migration increased; endothelial markers decreased (CD31, VE-cadherin, CD34, FVIII); mesenchymal markers increased (α-SMA, S100A4, COL1A1, SM22α, N-cadherin)
Marjorie Adjuto-Saccone et al., 2021 [40]	Recombinant TNF-α	20, 50, 100 ng/mL TNF-α (mainly 100 ng/mL) for 24–168 h (CD31 decreased from 24 h; α-SMA increased from 48 h)	Spindle-like morphology; migration increased; angiogenesis decreased; endothelial markers decreased (CD31, VE-cadherin, CD34); mesenchymal markers increased (α-SMA, S100A4, COL1A1, SM22α, N-cadherin)
Choi et al., 2016 [41]	HSPB1 si/shRNA	siRNA 1–3 days; nasal delivery of Hspb1 shRNA after 2 weeks of tumor induction, analyzed at 14 weeks	Endothelial markers decreased (VE-cadherin, CD31); mesenchymal marker α-SMA increased; VEGF-driven tube formation decreased; in vivo α-SMA^+^/CD31^+^ cells and collagen deposition increased; TGF-β1 and FSP1 increased; tumor fibrosis and progression increased
Choi et al., 2018 [28]	Irradiation	Single 20 Gy dose of radiation; assessed 1–23 days post-irradiation	α-SMA^+^/CD31^+^ partial EndMT increased; collagen deposition increased; tumor growth and metastasis increased; CD44v6^+^ cancer stem-like cells increased; α-SMA^+^/NG2^+^ pericytes increased; lineage-traced EndMT in tdTomato Ecs
Paola Gasperini et al., 2012 [43]	KSHV infection	Infection with rKSHV.219 (2.5 mL supernatant of virus-producing VERO cells), 12-day culture with puromycin selection	Actin reorganization; migration increased; endothelial markers decreased (CD31, VE-cadherin, Tie2, CD34); mesenchymal markers increased (CD146, NG-2, vimentin, PDGFRβ, α-SMA); in KS lesions, LANA^+^ cells showed low CD31 and high mesenchymal markers

*Abbreviations:* ACM, activated conditioned medium; CAF, cancer-associated fibroblast; CM, conditioned medium; EAC, esophageal adenocarcinoma; EC, endothelial cell; EndMT, endothelial-to-mesenchymal transition; HDLEC, human dermal lymphatic endothelial cells; HEMEC, human esophageal microvascular endothelial cells; HMEC-1, human microvascular endothelial cells; HUVEC, human umbilical vein endothelial cells; IFF, 3D microfluidic device with interstitial fluid flow; KSHV, Kaposi’s sarcoma-associated herpesvirus; KS, Kaposi’s sarcoma; lncRNA, long non-coding RNA; OPN, osteopontin; ROS, reactive oxygen species; shRNA, short hairpin RNA; siRNA, small interfering RNA; TEER, transendothelial electrical resistance; TME, tumor microenvironment.

**Table 2 biology-14-01403-t002:** Summary of transcriptomic/proteomic evidence for cancer-induced EndMT (condensed): markers for EndMT cell-type identification and key phenotypes. Columns included: Reference; Sequencing type; EndMT distinguishing markers; EndMT key phenotype (including impacts on cancer and the TME). The full dataset is provided in Appendix A.

Reference	Sequencing Type	EndMT Distinguishing Markers	EndMT Key Phenotype
Han Luo et al., 2022 [44]	scRNA-seq	CAF subtype co-expressing endothelial markers (e.g., vWF, PLVAP) and mesenchymal markers (e.g., ACTA2, RGS5) designated as CAFEndMT; Endothelial Cell-Specific Molecule 1 (ESM1) proposed as a CAFEndMT-specific marker.	TEC, CAFEndMT, and CAFmyo continuity suggested by trajectory analysis; CAFEndMT shows a high angiogenesis hallmark signature; in some cancers, patients with high CAFEndMT gene-set signatures exhibit poorer survival; strong predicted interaction with SPP1^+^ TAMs.
Liu Q. et al., 2025 [21]	scRNA-seq, ST-seq, bulk RNA-seq	PECAM1^+^ endothelial cluster with high expression of mesenchymal markers (ACTA2, COL1A1, RGS5) and enrichment of matrix-related pathway gene sets designated as COL1A1^+^ EC; differentially expressed genes with significant increases used to define an EndMT signature (EdMTS).	EdMTS is higher in GC than in adjacent normal tissue, higher in advanced GC, and highest in liver metastasis; strongly positively correlated with cancer-related processes (hypoxia, invasion, migration, TGF-β signaling); contributes to immune suppression/evasion; associated with poorer survival.
Minghui Hou et al., 2025 [45]	scRNA-seq, bulk RNA-seq	After primary classification by endothelial markers (CD34, PECAM1, VWF), a secondary CD90-positive mesenchymal pattern was used to designate the cluster SAEndo2; ESM1 selectively expressed.	In SAEndo2, mesenchymal/ECM-related pathways (ECM remodeling, EMT) are upregulated; CD34^+^CD90^+^ ECs display high expression of mesenchymal markers (FN1, α-SMA, COL4A2, MMP-9, PDGF) and EndMT-related transcription factors (TWIST, SLUG, SNAI); linked to poor survival and adverse prognosis; promotes tumor-cell migration and invasion.
Li Ji et al., 2025 [46]	ST-seq, bulk RNA-seq, LFQ proteomics	No explicit EndMT cluster defined; within fibroblast subsets, an rGBM-enriched cluster with high ECM Gene Ontology enrichment and high CAF markers (COL3A1, COL1A1, COL1A2, FN1) designated as a CAF-enriched cluster.	In HCMECs, endothelial markers (VE-cadherin, CD31) decrease, mesenchymal markers (α-SMA, COL1A1, FN1) increase, TGF-β is upregulated, and drug resistance is implicated.

*Abbreviations:* CAF, cancer-associated fibroblast; CAFEndMT, EndMT-like CAF subtype; CAFmyo, cancer-associated myofibroblast; EC, endothelial cell; ECM, extracellular matrix; EdMTS, EndMT signature; EMT, epithelial-to-mesenchymal transition; EndMT, endothelial-to-mesenchymal transition; GC, gastric cancer; HCMEC, human cerebral microvascular endothelial cells; LFQ, label-free quantitation; rGBM, recurrent glioblastoma; SAEndo2, scar-associated endothelial 2; scRNA-seq, single-cell RNA sequencing; ST-seq, spatial transcriptomics sequencing; TAM, tumor-associated macrophages; TEC, tumor endothelial cell; TME, tumor microenvironment.

## Data Availability

This study is a review of previously published literature. No new primary data were generated. The figures and tables presented summarize data extracted from published sources, as cited in the reference list, and additional details are available in the Appendix A.

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
