# Peer review of "Cancer-Associated Fibroblasts Arising from Endothelial-to-Mesenchymal Transition: Induction Factors, Functional Roles, and Transcriptomic Evidence"

_biology, 2025, doi:10.3390/biology14101403_

Round 1

Reviewer 1 Report

Comments and Suggestions for Authors

The manuscript by Han et al. is a comprehensive review of the evidence supporting the origin of  a subset of cancer associated fibroblasts (CAFs) coming from endothelial cells that have undergone or partially undergone endothelial to mesenchymal transition (EndMT).  The manuscript is well organized and up to dat on current literature.  The only thing I noted was I felt section 2.6.2 on radiation induction of CAFs was a little confusing due to the apparent paradox that deletion of TGFBR2 deletion would enhance EndMT when TGFbs generally increase EndMT.  A statement either noting the paradox or potentially why this would still be consistent with previous evidence of TGFB ligands inducing EndMT would be useful.

Author Response

We sincerely appreciate your review of our manuscript. Our responses to your comments are provided below.

Comment 1: The manuscript by Han et al. is a comprehensive review of the evidence supporting the origin of  a subset of cancer associated fibroblasts (CAFs) coming from endothelial cells that have undergone or partially undergone endothelial to mesenchymal transition (EndMT).  The manuscript is well organized and up to dat on current literature.  The only thing I noted was I felt section 2.6.2 on radiation induction of CAFs was a little confusing due to the apparent paradox that deletion of TGFBR2 deletion would enhance EndMT when TGFbs generally increase EndMT.  A statement either noting the paradox or potentially why this would still be consistent with previous evidence of TGFB ligands inducing EndMT would be useful.

Response 1: 

In response to the reviewer’s comments, we added a clarifying sentence to Section 2.6.2 (page 10-11) explaining how the observation can still be consistent with prior evidence. The content is as follows:

However, given that TGF-β signaling is a principal driver of EndMT, the increase in EndMT observed with TGFBR2 reduction may seem paradoxical. In this study, the au-thors showed—both in vitro and in vivo—that endothelial cell–specific Tgfbr2 knockdown led to increased irradiation-induced p-SMAD2/3, and they proposed that loss of TGFBR2 can compensatorily amplify SMAD2/3 signaling via TGFBR1. Accordingly, even with re-duced TGFBR2, the TGF-β pathway remains sufficiently activated to drive EndMT.

Reviewer 2 Report

Comments and Suggestions for Authors

1) The article title would have been better without the abbreviations.
2) The paper lacks illustrative material in the form of diagrams. I recommend the authors carefully consider expanding the graphical presentation of their materials.
3) The authors mention the heterogeneity of CAFs, but do not always critically evaluate whether EndMT cells can be clearly classified as pro- or anti-tumor subtypes.
4) The review lists numerous signaling pathways (TGF-β, NOTCH, ROS, etc.), but does not provide a comparative analysis of their contribution to different types of cancer.
5) A section on potential therapeutic strategies targeting EndMT and their limitations is missing.

Author Response

We sincerely appreciate your review of our manuscript. Our responses to your comments are provided below.

Comments 1: The article title would have been better without the abbreviations.

Response 1: In response to the reviewer’s comments, we revised the manuscript to remove abbreviations (page 1). 

Title: Cancer-Associated Fibroblasts Arising from Endothelial-to-Mesenchymal Transition: Induction Factors, Functional Roles, and Transcriptomic Evidence

Comments 2: The paper lacks illustrative material in the form of diagrams. I recommend the authors carefully consider expanding the graphical presentation of their materials.

Response 2: In response to the reviewer’s comments, we added the figure (provided as a Word file) near the beginning of Section 2. EndMT Induction in Cancer: Models and Factors (page 3-4). By inserting Figure 2, our aim is to provide a brief, upfront overview of the induction factors and to facilitate understanding of Table 2, which summarizes Section 2. We felt that presenting Table 2 alone could make it difficult to grasp the content at a glance, so adding Figure 2 improves accessibility and comprehension of the table. 
For Section 3, the content corresponding to the table is relatively concise compared with Table 2; therefore, we considered the table alone sufficient for clear and straightforward understanding and did not prepare an additional figure.

Comments 3: The authors mention the heterogeneity of CAFs, but do not always critically evaluate whether EndMT cells can be clearly classified as pro- or anti-tumor subtypes.

Response 3: We accepted the reviewer’s suggestion and, in the Conclusion section (page 18), we explicitly evaluated EndMT-derived CAFs as a pro-tumor subtype, as follows: 

Additionally, the EndMT-derived CAFs highlighted in this review exhibited pro-tumor characteristics. Given the heterogeneity of CAF subtypes—some of which can be pro- or anti-tumor—EndMT-derived CAFs to date appear to align predominantly with pro-tumor features, while exceptions cannot be excluded.

However, not all studies included in our review directly assessed the impact of EndMT on cancer; therefore, we did not classify each individual paper by pro-/anti-tumor subtype. Moreover, even in studies that did examine cancer impact, authors often clearly described roles in promoting tumor progression without explicitly labeling them as ‘pro-tumor,’ which we consider sufficient to interpret these findings accordingly. In this revision, we therefore updated the Conclusion to state explicitly that EndMT-derived CAFs predominantly align with pro-tumor features and, on this basis, we propose EndMT-derived CAFs as a pro-tumor subtype in this review.

Comments 4: The review lists numerous signaling pathways (TGF-β, NOTCH, ROS, etc.), but does not provide a comparative analysis of their contribution to different types of cancer.

Response 4: We covered approximately 14 different cancer types in this review. Some were discussed in multiple studies, whereas others were not. Accordingly, we judged that the number of repeated instances was insufficient to characterize signaling pathways as cancer type–specific. In addition, most of the studies we surveyed did not explicitly relate their findings to tumor-type specificity; therefore, we did not analyze contributions by cancer type.

Comments 5:  A section on potential therapeutic strategies targeting EndMT and their limitations is missing.

Response 5: We introduced studies related to therapies targeting EndMT. For example, we cited the work of Se-Hyuk Kim et al. (2019), which improved drug resistance by inhibiting EndMT through GSK-3β suppression, and Li Ji et al. (2025), which overcame drug resistance by targeting TNC and FLNC, known inducers of EndMT. However, we did not add a separate section on therapeutic strategies and limitations because our primary aim was to introduce individual EndMT studies. Rather than creating a standalone ‘Therapeutic Strategies’ section, we preferred to present therapy-related content within the context of each study. Accordingly, we discussed EndMT-targeted therapeutic approaches in the studies mentioned above without separating them into an independent section. In addition, because relatively few of the included papers addressed explicit therapeutic strategies, we did not provide a dedicated discussion of limitations specific to such strategies.

Reviewer 3 Report

Comments and Suggestions for Authors

The review presented is a work of great interest, given the growing attention currently being paid to the topic of EndMT and the increasingly clear role of CAFs. The excursus provided by the authors highlights the main molecular discoveries and the tools used, within a broad timeline. I believe that the contribution this review makes to the historical and scientific reconstruction of the role of CAFs and EndMT is both timely and necessary. Overall, the work is well executed, clear, and supported by coherent and reliable references. The presence of tables and images helps the reader to synthesize the information.

The purpose of the review and the need to define the characteristics of EndMT derived from CAFs have been fully achieved.

My only suggestion is to emphasize instances where similar mechanisms or overlapping pathways have been identified across different studies or experimental models. Highlighting these commonalities would enhance the utility of the review, particularly for researchers aiming to identify and investigate molecular targets involved in regulating this process.

Author Response

We sincerely appreciate your review of our manuscript. Our responses to your comments are provided below.

Comments 1: My only suggestion is to emphasize instances where similar mechanisms or overlapping pathways have been identified across different studies or experimental models. Highlighting these commonalities would enhance the utility of the review, particularly for researchers aiming to identify and investigate molecular targets involved in regulating this process.

Response 1: In response to the reviewer’s comments, we emphasized in the Conclusion section (page 18) the factors—recurrently reported among the mechanisms introduced in this review—that could be considered potential targets for EndMT, as follows:
Mechanistically, reported drivers of EndMT include TGF-β signaling, cytoskeletal/mechanical stress, ROS, OPN, PAI-1, IL-1β, GSK-3β, HSP90α, TIE1, TNF-α, HSPB1, and NOTCH signaling. Notably, TGF-β signaling has been recurrently observed across multiple studies and models, underscoring its central role in EndMT induction. In addition, factors such as OPN, PAI-1, and ESM1 have been repeatedly reported, suggesting that they represent important targets in EndMT biology. In particular, OPN has been implicated in EndMT across diverse cancer types and is highly expressed by multiple cell populations within the TME; taken together, these observations nominate OPN as a potentially critical target for future investigations of cancer-associated EndMT.

Round 2

Reviewer 2 Report

Comments and Suggestions for Authors The authors of the cautious article may be accepted in its current form.